# Accelerated cell divisions drive the outgrowth of the regenerating spinal cord in axolotls

**Fabian Rost[1†], Aida Rodrigo Albors[2,3†§], Vladimir Mazurov[2,3], Lutz Brusch[1,4], Andreas Deutsch[1,4], Elly M Tanaka[2,3*‡], Osvaldo Chara[1,5*‡]**

[1]Center for Information Services and High Performance Computing, Technische Universität Dresden, Dresden, Germany; [2]Deutsche Forschungsgemeinschaft – Center for Regenerative Therapies Dresden, Dresden, Germany; [3]Max Planck Institute of Molecular Cell Biology and Genetics, Dresden, Germany; [4]Center for Advancing Electronics Dresden, Dresden, Germany; [5]Systems Biology Group, Institute of Physics of Liquids and Biological Systems, National Scientific and Technical Research Council, University of La Plata, La Plata, Argentina

**\*For correspondence:** elly.
tanaka@crt-dresden.de (EMT);
osvaldo.chara@tu-dresden.de
(OC)

[†]These authors also contributed
equally to this work
[‡]These authors also contributed
equally to this work

**Present address:** [§]Division of
Cell and Developmental Biology,
School of Life Sciences,
University of Dundee, Dundee,
United Kingdom

**Competing interests:** The
authors declare that no
competing interests exist.

**Reviewing editor:** Marianne
Bronner, California Institute of
Technology, United States

**Abstract** Axolotls are unique in their ability to regenerate the spinal cord. However, the mechanisms that underlie this phenomenon remain poorly understood. Previously, we showed that regenerating stem cells in the axolotl spinal cord revert to a molecular state resembling embryonic neuroepithelial cells and functionally acquire rapid proliferative divisions (*Rodrigo Albors et al., 2015*). Here, we refine the analysis of cell proliferation in space and time and identify a high-proliferation zone in the regenerating spinal cord that shifts posteriorly over time. By tracking sparsely-labeled cells, we also quantify cell influx into the regenerate. Taking a mathematical modeling approach, we integrate these quantitative datasets of cell proliferation, neural stem cell activation and cell influx, to predict regenerative tissue outgrowth. Our model shows that while cell influx and neural stem cell activation play a minor role, the acceleration of the cell cycle is the major driver of regenerative spinal cord outgrowth in axolotls.

## Introduction

Neural stem cells exist in the spinal cord of all vertebrates, but only in salamanders these cells are mobilized efficiently to resolve spinal cord injuries (*Becker and Becker, 2015*; *Tanaka and Ferretti, 2009*). In axolotls, this is best exemplified following tail amputation, when cells adjacent to the cut end regrow a fully functional spinal cord (*Holtzer, 1956*; *Mchedlishvili et al., 2007*). Despite the regenerative potential of axolotl neural stem cells, little was known about the molecular changes occurring upon them and the changes in cell behavior that lead to the rapid expansion of the stem cell pool during regeneration.

In our previous study, we looked at spinal cord regeneration at the molecular and cellular levels. There, we found that resident SOX2[+] neural stem cells re-activate an embryonic-like gene expression program following tail amputation (*Rodrigo Albors et al., 2015*). Part of this program involves the re-establishment of planar cell polarity (PCP) signaling, the downregulation of pro-neural genes, and upregulation of proliferation-promoting genes. In line with these gene expression changes, we also found that regenerating neural stem cells speed up their cell cycle, and switch from neuron-generating to proliferative cell divisions. PCP turned out to be key for the efficient and orderly expansion of the regenerating spinal cord, at least in part by instructing cells to divide along the growing axis. However, besides oriented cell division, whether other cellular mechanisms such as convergence and

extension, which leads to the narrowing and lengthening of tissues, are involved in the rapid expansion of the regenerating spinal cord remained unknown.

In this follow-up study we investigate the contribution of different cellular mechanisms to the elongation of the regenerating spinal cord in the axolotl. To address this question, we apply a quantitative modeling approach to causally link previous (*Rodrigo Albors et al., 2015*) and new datasets to the time-course of spinal cord outgrowth. In particular, we calculate neural stem cell density from previous measurements (*Rodrigo Albors et al., 2015*) to show that convergence and extension is negligible. We make use of cell proliferation-related measurements along the anterior-posterior axis (AP) of the spinal cord (*Rodrigo Albors et al., 2015*) to identify a high-proliferation zone, which initially extends 800 μm anterior to the amputation plane, and calculate changes in cell cycle kinetics within this zone. By tracing sparsely-labelled cells, we also determine the cell influx into the regenerating spinal cord. Finally, we set up a mathematical model of spinal cord outgrowth that incorporates cell proliferation, neural stem cell activation, and cell influx. Using this model, we test the contribution of each of these cellular mechanisms to the regenerative spinal cord outgrowth. Comparing the model predictions with experimental data of tissue outgrowth we show that while cell influx and activation of quiescent neural stem cells play a minor role, the acceleration of the cell cycle in the high-proliferation zone is the major driver of the observed regenerative spinal cord outgrowth.

## Results

### The regenerating spinal cord grows with increasing velocity

To refine the outgrowth time-course of the regenerating spinal cord, we measured the spinal cord outgrowth in individual axolotls, 2–3 cm snout to tail, during the first 8 days of regeneration (*Figure 1A*, *Figure 1—figure supplement 1* and *Rost et al., 2016b*). Initially, the regenerating spinal cord extended slowly to a mean outgrowth of 0.45 ± 0.04 mm at day 4 (*Figure 1B*). Thereafter, the spinal cord grew faster, reaching an outgrowth of 2.26 ± 0.07 mm by day 8.

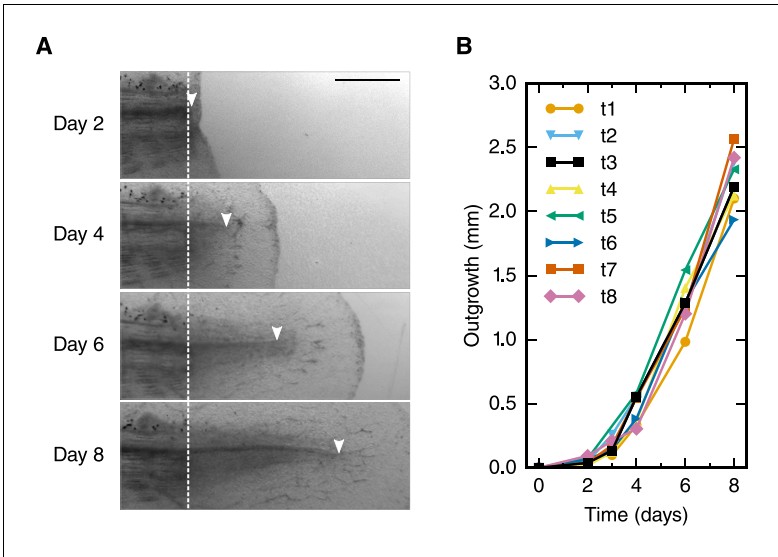

**Figure 1.** Spinal cord outgrowth time-course during regeneration. (A) Representative images of a regenerating spinal cord after tail amputation (individual time-lapse images are in *Figure 1—figure supplement 1*). The white dashed line marks the amputation plane. The arrowheads mark the tip of the regenerating spinal cord. Scale bar, 1 mm. (B) Spinal cord outgrowth time-course during the first eight days after amputation (n = 8 axolotls).

The following figure supplement is available for figure 1:

**Figure supplement 1.** Images used for spinal cord outgrowth measurements in *Figure 1B*.

## The density of neural stem cells stays constant along the AP axis of the regenerating spinal cord

To explain the outgrowth time-course of the regenerating spinal cord in terms of underlying cellular mechanisms, we first set out to translate tissue outgrowth into cell numbers. To quantitatively investigate neural stem cell arrangement in space and time, we revisited our previously published dataset of the number of SOX2$^+$ cells per cross section in uninjured and regenerating spinal cords (*Figure 2A* and see Materials and methods) (*Rodrigo Albors et al., 2015*). We found that the number of SOX2$^+$ cells per spinal cord cross section is constant along the AP axis in both uninjured and regenerating samples at any time (*Figure 2B,B'* and *Figure 2—figure supplement 1* and see Materials and methods). We also found that the number of SOX2$^+$ cells per cross section spatially averaged along the AP axis is constant during regeneration (*Figure 2C* and see Materials and methods). On average, 30.4 ± 0.6 SOX2$^+$ cells make up the circumference of the axolotl spinal cord. Since the length of SOX2$^+$ cells along the AP axis does not change during regeneration ($l_c$ = 13.2 ± 0.1 μm) (*Rodrigo Albors et al., 2015*), the density of cells along the AP axis is spatially homogeneous and equal to 2.3 ± 0.6 cells/μm (*Figure 2A*).

Taken together, these findings allow us to exclude mechanisms such as cell shape changes and convergence and extension as driving forces of regenerative spinal cord outgrowth in the axolotl. Instead, constant neural stem cell density implies an increasing neural stem cell number during regeneration. This suggests that the expansion of the regenerating neural stem cell pool mostly relies on proliferation-based mechanisms.

## Cell proliferation increases within an 800 μm zone anterior to the amputation plane in four-day regenerates

To determine spatial and temporal changes in cell proliferation during regeneration, we calculated different cell proliferation parameters along uninjured and regenerating spinal cords. In our previous study, we quantified the number of proliferative cells, i.e. SOX2$^+$ cells that are positive for proliferating cell nuclear antigen (PCNA) and the number of cells in mitosis, i.e. SOX2$^+$/PCNA$^+$ cells with condensed chromosomes based on Hoechst DNA stain (*Rodrigo Albors et al., 2015*). Here, we used these datasets to estimate the growth fraction, i.e. the fraction of proliferative cells and the mitotic index, i.e. the ratio of mitotic cells over proliferative cells. Although neither SOX2$^+$/PCNA$^+$ cells nor mitotic cells showed any evident spatial pattern along the AP axis in uninjured animals (*Figure 2D*, points), they showed a tendency to increase posteriorly from day 4 (*Figure 2D'*, points). To elucidate whether proliferation was patterned along the AP axis during regeneration, we tested the data with a mathematical model of two spatially homogeneous zones characterized by their growth fraction and mitotic index and separated by a border that we call the *switchpoint* (*Figure 2E,E'*). We reasoned that in the absence of an AP pattern of cell proliferation the two zones would be indistinguishable; while if cell proliferation would be locally increased, the model would allow us to determine the magnitude and the location of the increased cell proliferation. For a given growth fraction and mitotic index, the model predicts the expected number of proliferative cells and mitotic cells per cross section (*Figure 2—figure supplement 2*). Hence, we fitted the model to the cell number datasets of uninjured and regenerating spinal cords at day 3, 4, 6 and 8 after amputation (*Figure 2D,D'*, *Figure 2—figure supplement 3* and *Figure 2—figure supplement 4*) to determine the growth fraction, the mitotic index, and the switchpoint for each time point (*Figure 2F–F''*). Not surprisingly, we found that in the uninjured spinal cord the growth fraction and the mitotic index in the two modeled zones are not significantly different (*Figure 2D,F,F'* and *Figure 2—figure supplement 3*). Similarly, at day 3 there are no significant differences between the two zones (*Figure 2F,F'* and *Figure 2—figure supplement 3*). In contrast, the growth fraction and the mitotic index are higher in the posterior zone from day 4 onward (*Figure 2D',F,F'* and *Figure 2—figure supplement 3*). These findings reveal that a high-proliferation zone emerges in the regenerating spinal cord at day 4. At this time point, the switchpoint between the two zones is located 800 ± 100 μm anterior to the amputation plane, but shows the tendency to shift posteriorly as the regenerating spinal cord grows (*Figure 2F''*).

Next, we combined the mitotic index measurements with our previous cell cycle length estimates (*Rodrigo Albors et al., 2015*) to establish how the proliferation rate changes during regeneration (*Figure 2G* and see Materials and methods). We find that the proliferation rate is 0.06 ± 0.02 per

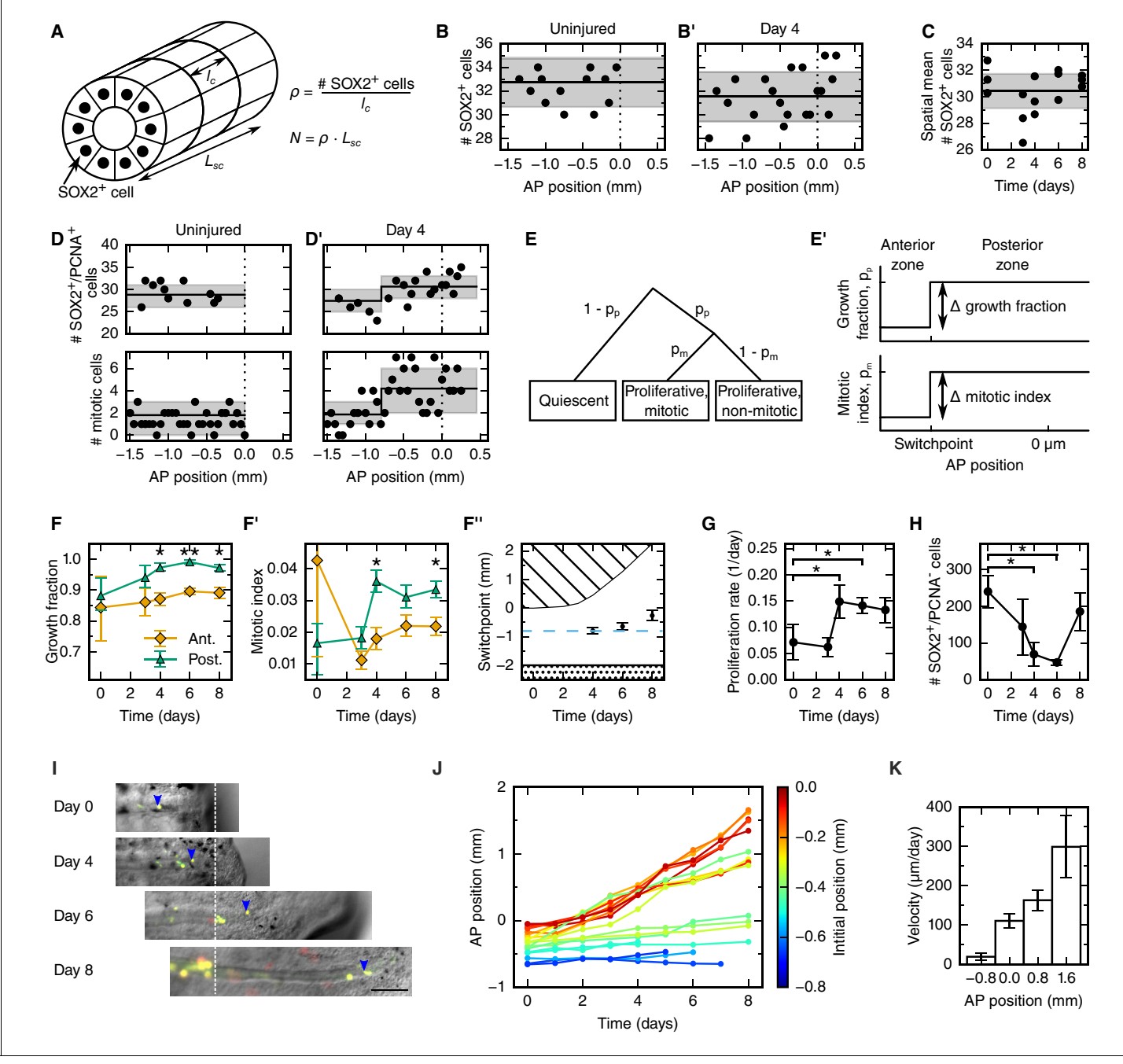

**Figure 2.** Cellular mechanisms underlying spinal cord outgrowth. (**A**) Sketch of measurements taken to estimate the density and total number of neural stem cells (nuclei, black dots) in the axolotl spinal cord. The density of SOX2$^+$ cells, $\rho$, is the ratio of the number of SOX2$^+$ cells per cross section (# stem cells) and the mean AP cell length, $l_c$. The density of SOX2$^+$ cells is the proportionality constant between the total number of stem cells in a zone along the spinal cord with zone length, $L_{SC}$. (**B,B'**) Number of SOX2$^+$ cells per cross section along the AP axis of a selected uninjured (**B**) and a selected day 4-regenerating spinal cord (**B'**). Black line and gray region indicate the mean number of SOX2$^+$ cells and the standard deviation, respectively. Plots for all individual axolotls in *Figure 2—figure supplement 1*. (**C**) Spatial mean of the number of SOX2$^+$ cells per cross section of individual axolotls against time (black dots). Black line and gray region indicate the mean number of SOX2$^+$ cells and the standard deviation of all animals, respectively. (**D,D'**) Number of SOX2$^+$/PCNA$^+$ cells per cross section (upper panel) and mitotic cells per section (lower panel) along the AP axis in a selected uninjured (**D**) and a selected day 4-regenerating spinal cord (**D'**). Black line and the gray region show the expected number and the 68% confidence belt for the best fit of the model with two spatial proliferation zones, respectively. Plots for all animals in *Figure 2—figure supplement 3*. (**E**) Possible cell states in the two spatial proliferation zones model used to analyze the spatial cell proliferation dataset (**D,D'**). $p_p$, probability that a cell is proliferative, otherwise quiescent. $p_m$, probability that a proliferative cell undergoes mitosis at the time of analysis. (**E'**) The model assumes two proliferation zones. The location of the border between zones is called *switchpoint*. (**F–F'**) Results of model fitting for growth fraction (**F**) and mitotic
*Figure 2 continued on next page*

*Figure 2 continued*

index time-course (F') in the anterior (orange diamonds) and posterior (green triangles) zone. Error bars indicate the 68% credibility interval. (F'') Black dots mark the switchpoint. Blue dashed line marks 800 μm anterior to the amputation plane. The dashed region marks the space outside of the embryo, the dotted region marks the unaffected part of the embryo. (G) Proliferation rate time-course in the high-proliferation zone. (H) Total number of SOX2$^+$/PCNA$^-$ cells in the high-proliferation zone (mean ± linearly propagated 1-σ error). (I) Selected time-lapse images of clone (blue arrowhead) tracking during spinal cord regeneration. Dashed line marks the amputation plane. Scale bar, 200 μm (J) Tracking of 19 clones along the AP axis during regeneration. Clone trajectories are color coded by their initial position. (K) Clone velocities at different positions along the AP axis.

The following figure supplements are available for figure 2:

**Figure supplement 1.** Number of SOX2$^+$ cells per cross section along the AP axis for all 15 animals.

**Figure supplement 2.** Simulation of the spatial model of cell counts to analyze the spatiotemporal pattern of cell proliferation.

**Figure supplement 3.** Number of SOX2$^+$/PCNA$^+$ cells per cross section (upper panel) and mitotic cells per section (lower panel) along the AP axis for all 15 animals.

**Figure supplement 4.** Posterior marginal distributions for the parameters of the spatial model of cell counts to analyze the spatiotemporal pattern of proliferation.

**Figure supplement 5.** Cell cycle length time-course calculated from the proliferation rate time-course shown in *Figure 2G*.

day in the uninjured spinal cord which corresponds to a cell cycle length of 10 ± 4 days (*Figure 2—figure supplement 5*). The proliferation rate is similar at day 3. However, at day 4 the proliferation rate increases to about 0.15 per day corresponding to a cell cycle length of about five days and the proliferation rate remains that high until day 8.

## Quiescent neural stem cells re-enter the cell cycle during regeneration

Two possible scenarios could lead to the observed increased growth fraction in the high-proliferation zone (*Figure 2F*): the activation of quiescent neural stem cells, or the dilution of quiescent cells by the expansion of the proliferating cell population. If quiescent cells were activated, the total number of quiescent cells in the high-proliferation zone would decrease. We estimated the total number of quiescent cells in the high-proliferation zone from the mean number of SOX2$^+$/PCNA$^-$ cells per cross section, the mean AP cell length, and the outgrowth time-course (see Materials and methods). The number of SOX2$^+$/PCNA$^-$ cells drops from 180 ± 30 at day 0 to 23 ± 13 at day 6 (*Figure 2H*) which suggests that quiescent SOX2$^+$ cells get activated and re-enter the cell cycle upon injury. The number of quiescent SOX2$^+$ cells appears to increase again at day 8, when cells resume neurogenesis (*Rodrigo Albors et al., 2015*).

## Cells translocate faster the closer they are to the tip of the regenerate

Cell movement could also contribute new cells to the regenerative spinal cord outgrowth. To investigate whether anterior spinal cord cells move into the high-proliferation zone, we followed individual cells during regeneration. For that, we electroporated cells with a dual fluorescent reporter plasmid (cytoplasmic GFP and nuclear mCherry) at very low concentration to achieve sparse labelling of cells and tracked them daily during the first 8 days of regeneration (*Figure 2I*). We found that labelled cells preserve their original spatial order: cells located close to the amputation plane end up at the posterior end of the regenerated spinal cord (*Figure 2J*). Most-anterior cells, however, almost do not change their position. From the clone trajectories, we calculated the mean clone velocity at different positions along the AP axis (*Figure 2K* and see Materials and methods). Clones initially located 800 μm anterior to the amputation plane translocate slowly, with a velocity of 20 ± 9 μm/day. In contrast, the more posterior a clone is, the faster it translocates (*Figure 2K*).

## Cell proliferation drives the outgrowth of the regenerating spinal cord

The fact that cell density along the AP axis is constant in space and time (*Figure 2B–C*) made us reason that the spinal cord must grow as a result of increasing cell numbers. In line with this, we found

a high-proliferation zone, first spanning from 800 µm anterior to the amputation plane, and showed that the increase in cell proliferation is due to both (i) the acceleration of the cell cycle and (ii) the activation of quiescent stem cells (*Figure 2D–H*). The influx of cells that we identified could also contribute to increasing cell numbers in the regenerating spinal cord (*Figure 2I–K*). To assess the contribution of these cellular mechanisms to the outgrowth time-course, we used a quantitative mathematical modeling framework (*Greulich and Simons, 2016*; *Rué and Martinez Arias, 2015*; *Oates et al., 2009*). We formalized the influence of each cellular mechanism on the total number of proliferative and quiescent SOX2[+] cells in the high-proliferation zone in a mathematical model of cell numbers (*Figure 3A*, see Materials and methods, *Equations 3 and 4*). As cell density along the AP axis is constant, the cell number is proportional to the AP length of the growing high-proliferation zone. Hence, we can transform the model of cell numbers into an equivalent model for the tissue geometry that predicts the spinal cord outgrowth, $L(t)$, and growth fraction, $GF(t)$ at time $t$:

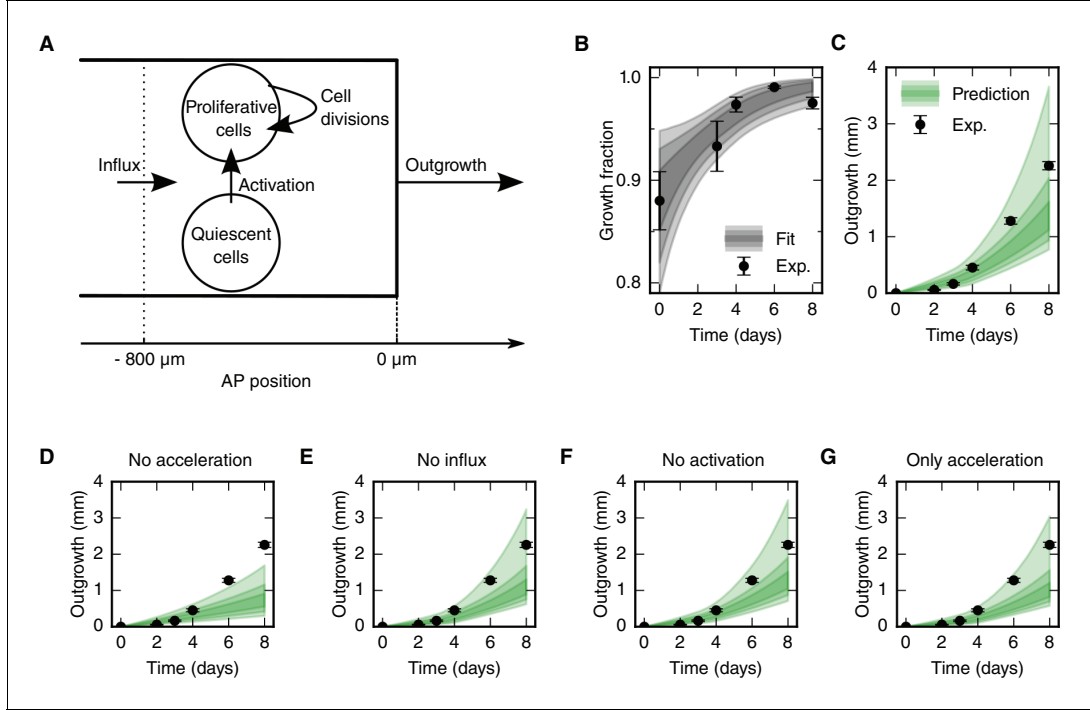

**Figure 3.** Mechanistic model of spinal cord outgrowth. (A) Sketch of cellular mechanisms included in the model: cell proliferation, quiescent cell activation, and cell influx into the 800 µm high-proliferation zone. (B) Growth fraction time-course of the SOX2[+] cell population in the high-proliferation zone as observed (black dots) and fitted by the model (gray shaded areas, from darker to lighter, 68%, 95% and 99.7% confidence intervals of the model prediction). (C) Spinal cord outgrowth during the first eight days of regeneration as observed (black dots, n = 8 axolotls) and predicted by the model (*Equations 1 and 2*) (green shaded areas, from darker to lighter, 68%, 95% and 99.7% confidence intervals). The model prediction is in agreement with the experimental data. (D–G) Prediction of spinal cord outgrowth for four model scenarios based on *Equations 1 and 2* with selected mechanisms switched off (green shaded areas). Black dots show the same experimental data as in panel (C). (D) The acceleration of the cell cycle is switched off. Hence, the proliferation rate is fixed to the basal proliferation rate of uninjured animals. (E) Cell influx is switched off ($v = 0$). (F) Quiescent cell activation is switched off ($k = 0$). (G) Cell influx and quiescent cell activation are switched off ($k = 0$, $v = 0$). Corresponding predictions for growth fraction in *Figure 3—figure supplement 1*.

The following figure supplements are available for figure 3:

**Figure supplement 1.** Prediction of the growth fraction in the high-proliferation zone for four model scenarios with selected mechanisms switched off (green shaded areas).

**Figure supplement 2.** Comparison of the spinal cord outgrowth prediction by our model with the measured outgrowth reported by *Fei et al. (2014)*.

$$\frac{dL(t)}{dt} = \overbrace{r(t)(L(t) + L_0)GF(t)}^{\text{divisions of proliferative cells}} + \overbrace{v}^{\text{influx of cells into the high-proliferation zone}} \qquad , L(t=0) = 0, \qquad (1)$$

$$\frac{dGF(t)}{dt} = \underbrace{(1 - GF(t))k}_{\text{activation of quiescent cells}} + \underbrace{(1 - GF(t))r(t)GF(t)}_{\text{dilution of quiescent cells in the expanding pool of proliferative cells}} \qquad , GF(t=0) = GF_0. \qquad (2)$$

where $L_0$ = 800 µm is the length of the high-proliferation zone, $GF_0$ is the growth fraction in uninjured tails, $r(t)$ is the proliferation rate at time $t$, $v$ is the velocity of cells 800 µm anterior to the amputation plane, and $k$ is the cell cycle entry rate. As we determined the proliferation rate time-course $r(t)$ (*Figure 2G*), the initial growth fraction $GF_0$ (*Figure 2F*) and the influx velocity $v$ (*Figure 2K*), only the cell cycle entry rate $k$ is unknown. By fitting the model to the experimental growth fraction data from day 0 to day 6 (*Figure 3B*), we determined this parameter as $k = 0.2 \pm 0.1$ day$^{-1}$. Importantly, the model predicts a spinal cord outgrowth time-course that recapitulates the observed experimental data (*Figure 3C*). This fit-free agreement shows that the acceleration of the cell cycle, the activation of quiescent neural stem cells, and an influx of cells into the regenerate quantitatively explain the observed spinal cord outgrowth.

To quantitatively determine the contribution of each cellular mechanism to the regenerative spinal cord outgrowth, we switched them off one by one in silico. First, we switched off the acceleration of the cell cycle, modeling growth only with basal cell proliferation, the influx of cells, and the activation of quiescent neural stem cells (*Figure 3D*). This predicted a maximum outgrowth of 1.7 mm (p=0.003) which is 0.6 mm shorter than the observed outgrowth at day 8. This result shows that the acceleration of the cell cycle is a key driver of regenerative spinal cord outgrowth. In contrast, switching off cell influx (*Figure 3E*) or the activation of quiescent neural stem cells (*Figure 3F*) has almost no effect on the predicted outgrowth, which suggests that these cellular mechanisms are not major drivers of regenerative spinal cord outgrowth. Indeed, even when we switched off both cell influx and cell activation the observed outgrowth time-course is in agreement with the model prediction (*Figure 3G*). Together, these results show that the acceleration of the cell cycle in cells that were already proliferating in the uninjured spinal cord can explain the observed spinal cord outgrowth during regeneration.

To test the prediction of our model against an independent experimental dataset, we revisited data of *Sox2*-knockout spinal cords (*Fei et al., 2014*). Fei and colleagues found evidences that *Sox2*-knockout prevented the acceleration of the cell cycle during regeneration and lead to shorter spinal cord outgrowth. In agreement with these findings, running our model with the acceleration of the cell cycle switched off recapitulated the shorter outgrowth in the *Sox2*-knockout condition (*Figure 3—figure supplement 2* and see Materials and methods).

## Discussion

The spinal cord tissue size and architecture is faithfully restored after tail amputation in axolotls. This unique regenerative capability relies on neural stem cells surrounding the central canal of the spinal cord. These cells re-activate an embryonic-like gene expression program that implements PCP signaling to make possible the increase in cell proliferation while maintaining a tube-like structure (*Rodrigo Albors et al., 2015*). However, the precise contribution of proliferation-based mechanisms to the outgrowth of the regenerated spinal cord and whether other cellular mechanisms are involved remained unknown.

Here, we combined detailed quantitative datasets with mathematical modeling to dissect the cellular mechanisms that underlie regenerative spinal cord outgrowth in axolotls. We found that the response to injury involves (i) changes in the cell proliferation rate, (ii) activation of quiescent neural stem cells, and (iii) cell influx into the regenerating spinal cord, while maintaining a surprisingly organized neural stem cell-scaffold. By modeling the contribution of each of these mechanisms to tissue outgrowth upon regeneration, we uncovered that the acceleration of the cell cycle is the main driver of regenerative spinal cord outgrowth in axolotls.

Increased proliferation of SOX2[+] cells upon spinal cord injury is a common feature among vertebrates (*Becker and Becker, 2015*). In zebrafish (*Hui et al., 2010, 2015*), *Xenopus* (*Gaete et al., 2012*), mouse (*Lacroix et al., 2014*) and axolotl (this work, *Rodrigo Albors et al., 2015*;

*Holtzer, 1956*) traumatic spinal cord injury triggers a long-range wave of increased cell proliferation. It is however clear that although the potential to replace lost cells or tissue exists in other species, they are not as efficient as axolotls at resolving spinal cord injuries. A more comprehensive characterization of cell proliferation responses is thus needed to understand fundamental differences between species with different regenerative capabilities. In our previous study, we uncovered that spinal cord stem cells in the axolotl speed up their cell cycle during regeneration (*Rodrigo Albors et al., 2015*). Performing detailed quantifications, we were now able to delineate a high-proliferation zone that initially spans from the 800 μm adjacent to the amputation plane to the regenerating tip, and later shifts posteriorly as the spinal cord regrows. Although some quiescent neural stem cells enter the cell cycle during regeneration, we demonstrate that the observed increase in proliferation is primarily due to the acceleration of the cell cycle within the regenerating neural stem cell pool. By performing experiments in silico using our mechanistic model of spinal cord regeneration, we demonstrate that the acceleration of the cell cycle can explain the observed spinal cord outgrowth.

We further applied our model to an independent experimental dataset in which *Sox2*-knockout spinal cords do not regrow properly upon amputation, due to the inability of *Sox2*-knockout cells to 'change gears' in response to injury (*Fei et al., 2014*). Indeed, *Sox2*-knockout cells express PCNA and are in theory able to proliferate, but their lower incorporation of the thymidine analog 5-ethynyl-2'-deoxyuridine (EdU) suggests that they cannot speed up the cell cycle (*Fei et al., 2014*). We were able to show that the reduced outgrowth in *Sox2*-knockout spinal cords can be quantitatively explained by the lack of cell cycle acceleration (*Figure 3—figure supplement 2*). However, it is important to point out that our model does not include the regulation of individual cellular mechanisms and thus it does not consider compensatory mechanisms that may operate under perturbed conditions. To apply our model to the *Sox2*-knockout dataset, we assumed that knocking out *Sox2* only affects the acceleration of the cell cycle. The fact that the model successfully recapitulated the experimental outgrowth in the *Sox2*-knockout scenario suggests that compensatory mechanisms might have a small contribution in this condition. Nevertheless, the validity of this assumption remains to be further investigated.

Our approach and findings highlight the importance of mathematical modeling and careful quantification of cellular mechanisms to understand the mechanisms of regeneration. Moreover, our detailed spatial and temporal characterization of cell proliferation may help to focus the search for key signals that might be operating in the high-proliferation zone to speed up the cell cycle of regenerative neural stem cells. It will be interesting to see whether the expression of AxMLP, the recently identified regeneration-initiating factor in axolotls (*Sugiura et al., 2016*), correlates in time and space with the high-proliferation zone. This work thus provides a deeper understanding of spinal cord regeneration in axolotls and new insights to help elucidating the molecular mechanisms that drive spontaneous spinal cord regeneration in vivo.

Besides the increase in cell proliferation, we uncovered an influx of cells into the regenerating spinal cord. Cells move along the AP axis of the spinal cord but maintain their relative position: cells translocate faster the closer they are to the amputation plane (*Figure 2J,K*). In line with earlier work (*Mchedlishvili et al., 2007*), we found that cells initially located within the 500 μm anterior to the amputation plane contribute to the regenerated spinal cord; while cells outside this zone translocate slower, and cells at 800 μm, the border of the high-proliferation zone, almost do not move. This would be consistent with a model in which cells are passively displaced, pushed by more anterior dividing cells. In this model, the more posterior a cell is the more cells anterior to that cell divide and the stronger is the push, making the cell translocate faster (*Figure 4*). Importantly, the proliferative response extends beyond the 500 μm anterior to the amputation plane that gives rise to the regenerated spinal cord (*Mchedlishvili et al., 2007*). In the light of this model, it is plausible that cells in the posterior 500 μm of the high-proliferation zone regenerate the spinal cord while cells from the anterior 300 μm of the high-proliferation zone replenish and push out the 500 μm regeneration source zone.

A notable finding of this study is that the increase in cell numbers during regeneration is tightly regulated so that the regenerating spinal cord extends while maintaining constant cell density and proper tube-like structure. This tube-like structure made up almost entirely of neural stem cells might be essential to act as a scaffold for rebuilding the spinal cord tissue architecture. Previously, we showed that the activation of PCP signaling within the source zone instructs cells to divide along the growing axis of the spinal cord and is key for effective spinal cord regeneration. This work

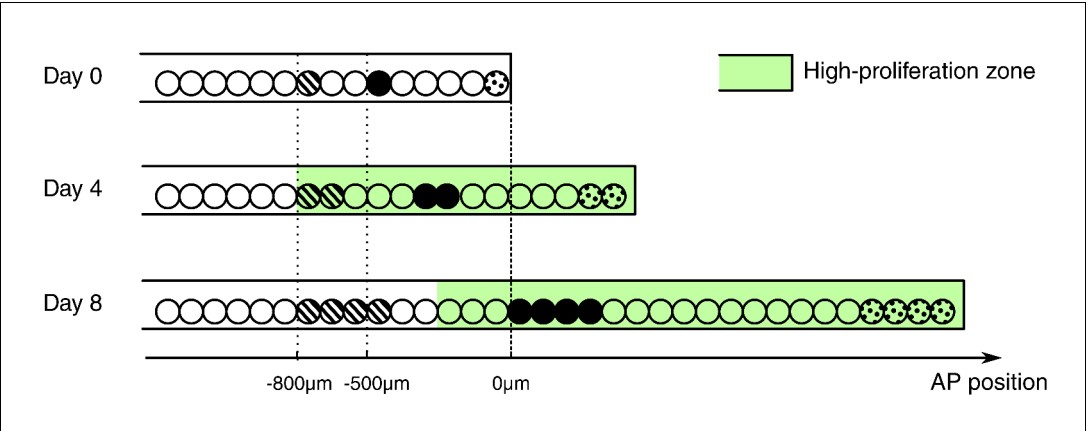

**Figure 4.** Conceptual model of spinal cord growth during regeneration. Only one row of stem cells is shown as circles and three cell clones are marked with different patterns (striped, black and dotted). In the uninjured spinal cord (Day 0), cells divide at a slow, basal proliferation rate (white background). From day 4 after amputation, cells speed up their cell cycle and the growth fraction increases, within a high-proliferation zone that initially extends 800 µm anterior to the amputation plane (green background). The density of neural stem cells along the spinal cord stays constant and spinal cord outgrowth is achieved by an increase in the total number of neural stem cells. Acceleration of the cell cycle in the high-proliferation zone is the major driver of this increase in cell numbers. Dividing cells might push cells posteriorly. The more posterior a cell is the more cells anterior to that cell divide and push the cell making it move faster: While an anterior clone (striped) hardly moves, clones in the center of the high proliferation zone (black) move faster. Clones that start at the amputation plane (dotted) stay at the tip of the regenerating spinal cord and move fastest.

highlights the importance of orderly and directed expansion of the neural stem cell pool for efficient spinal cord regeneration.

Together, our findings provide a quantitative mechanistic understanding of the cellular mechanisms that drive complete spinal cord regeneration in axolotls. By performing a quantitative modeling approach combined with quantitative experimental data, we found that axolotl spinal cord outgrowth is driven by the acceleration of the cell cycle in a pool of SOX2[+] neural stem cells restricted in space and time. Whether this peculiar spatiotemporal proliferative pattern is unique to the axolotl and how this correlates with injury-induced signals remain to be elucidated.

## Materials and methods

### Axolotls

Axolotls, *Ambystoma mexicanum*, from 2–3 cm in length snout-to-tail were used for experiments. Axolotls were kept in tap water in individual cups and fed daily with Artemia. Before any manipulation or imaging, axolotls were anaesthetized in 0.01% benzocaine. The axolotl animal work was performed under permission granted in animal license number DD24-9168.11-1/2012–13 conferred by the Animal Welfare Commission of the State of Saxony, Germany (Landesdirektion Sachsen).

### Measurement of spinal cord outgrowth

Images of regenerating tails were acquired on an Olympus SZX16 stereomicroscope using the CellF software by Olympus. Spinal cord outgrowth was measured from bright field images in Fiji (RRID: SCR_002285). First, the amputation plane which is clearly visible in the myotome was marked with a line. Then, the length between the intersection of the amputation plane with the spinal cord and the spinal cord tip was measured with Fiji's line tool.

### Cell count data

The cell count data of SOX2[+] and SOX2[+]/PCNA[+] cells per cross section and mitotic cells in 50 µm sections were taken from *Rodrigo Albors et al. (2015)*.

## Analysis of SOX2$^+$ cell count data

To test whether the SOX2$^+$ cells per cross section showed a spatial pattern along the AP axis or not, we used three different methods (*Figure 2B,B'*, *Figure 2—figure supplement 1*). First, it was tested if the cell count data linearly depends on spatial position along the AP axis using Bayesian inference (see 'Constant density' in *Rost et al., 2016a*). The slope was always smaller than 0.13 cells/mm and only significantly different from 0 (p<0.05) for 4 of the 15 replicates. Second, a model of two spatially homogeneous zones was fitted to the data using Bayesian inference (see 'Constant density' in *Rost et al., 2016a*). Here, only 4 of the 15 replicates showed a significant difference in density between the two zones (p<0.05). These first two methods indicated that, for an average animal, there is no significant change of the number of SOX2$^+$ cells per cross section along the AP axis. Third, the data was collapsed ignoring the spatial position, and the resulting cell count histogram was tested for being a normal distribution using the SciPy function scipy.stats.normaltest (*D'Agostino, 1971*; *D'Agostino and Pearson, 1973*). Only for one of the replicates the null hypothesis could be rejected (p<0.05), hence the SOX2$^+$ cell density in an average animal was considered spatially homogeneous with Gaussian noise in this study.

For each replicate the mean number of SOX2$^+$ cells per cross section averaged over all measurements along the AP axis was calculated. To access whether there was a significant change in this mean number, the replicates were grouped according to their time post amputation. A one-way ANOVA-test showed no significant differences among the groups (p=0.08, see 'Constant density' in *Rost et al., 2016a*).

## Analysis of proliferation count data

The counts of SOX2$^+$ cells, SOX2$^+$/PCNA$^+$ cells and mitotic cells were analyzed by fitting a mathematical model of two adjacent spatial proliferation zones to the data of each time point (*Figure 2D, D'*, *Figure 2—figure supplement 3*).

The model that predicts the number of SOX2$^+$/PCNA$^+$ cells per cross section and the number of mitotic cells in three-dimensional (3D) 50 μm sections based on the growth fraction and mitotic index was defined as follows: If the number of SOX2$^+$ cells for a specific cross section along the AP axis, $N_S$, had been measured, it was used for this section. If the data for the specific section was missing, $N_S$ was computed by assuming that there is a constant expected number of SOX2$^+$ cells per cross section and that the deviations from the expected value follow a normal distribution. The mean and standard deviation of this normal distribution were estimated by the sample mean and standard deviation of the sample of the measured numbers of SOX2$^+$ cells per cross section for each replicate. The number of SOX2$^+$cells in a cross section is independent from other cross sections. The state 'Proliferative', i.e. SOX2$^+$/PCNA$^+$, is independently assigned to each SOX2$^+$ cell with probability $p_P$ or 'Quiescent' with probability $1 - p_P$ (*Figure 2E*). Hence, for a given number of SOX2$^+$ cells in a cross section, $N_S$, the number of SOX2$^+$/PCNA$^+$ cells per cross section, $N_P$, follows a binomial distribution with $N_s$ experiments and success probability $p_P$. Consequently, the expected growth fraction equals $p_P$. As the number of mitotic cells, $N_M$, in 3D 50 μm sections was measured previously, we estimated the number of SOX2$^+$/PCNA$^+$ cells also in a 3D 50 μm section, $N_{PS} = 50~\mu m/l_{cell}~\cdot N_P$, where $l_{cell} = 13.2 \pm 0.1~\mu m$ is the mean AP length of SOX2$^+$ cells (*Rodrigo Albors et al., 2015*). Assuming that the cell cycle position and hence the cell cycle phase of each cell is independent of all other cells, the state 'Proliferative, mitotic' is independently assigned to each SOX2$^+$/PCNA$^+$ cell with probability $p_m$ or 'Proliferative, non-mitotic' with probability $1 - p_m$. Hence, the number of mitotic cells per section, $N_M$, follows a binomial distribution with $N_{PS}$ experiments and success probability $p_m$. Consequently, the expected mitotic index equals $p_m$. For given values of $p_P$ and $p_m$ the model gives a likelihood for the observed number of SOX2$^+$/PCNA$^+$ cells per cross section and mitotic cells per 3D section that can be used to fit the model parameters. To reflect the assumption of two spatial proliferation zones, $p_P$ and $p_m$ have spatial dependencies in the form of step functions (*Figure 2E'*). Hence, there can be different growth fractions and mitotic indices for the anterior and the posterior zone, respectively. The spatial position of the border between the zones is another model parameter termed *switchpoint*. Furthermore, variability between replicates in the switchpoint is modeled as a normal distribution with standard deviation $\sigma_{switch}$. Likewise, variability in growth fraction and mitotic index between replicates is modeled with a normal distribution with spatially homogeneous standard deviations $\sigma_{GF}$ and $\sigma_{mi}$, respectively. Hence, the resulting model to describe

the cell count data of all replicates at a given time point has eight parameters: the switchpoint, growth fraction and mitotic index in the anterior zone and in the posterior zone, respectively, and the inter-replicate variabilities $\sigma_{switch}$, $\sigma_{GF}$ and $\sigma_{mi}$. Those parameters were estimated with Bayesian inference using uniform priors for uninjured animals and at 3, 4, 6 and 8 days. Fitting was performed using a Markov chain Monte Carlo algorithm implemented in pymc (*Figure 2F–F''*, *Figure 2—figure supplement 4*, see also 'step_model_fixed_density_fit_per_timepoint' in *Rost et al., 2016a*). To verify the fitting procedure, test data were created by simulating our model with picked parameter values. These 'true' parameter values were then found to be included in the 95% credibility intervals of the parameter values inferred from the test data with our fitting procedure.

## Proliferation rate time-course

The cell cycle length at day six was estimated previously using a cumulative 5-bromo-2'-deoxyuridine (BrdU) labelling approach (*Rodrigo Albors et al., 2015*). For the sake of consistent methodology within the present study, the data were reanalyzed with bootstrapping using case resampling (see 'brdu_bootstrapping_day6' in *Rost et al., 2016a*). In agreement with the previous analysis the cell cycle length was estimated as 117 ± 12 hours corresponding to a proliferation rate of 0.21 ± 0.02 per day at six days after amputation.

As the mitotic index is proportional to the proliferation rate (*Smith and Dendy, 1962*), the mitotic index time-course in the high-proliferation zone was rescaled with the proliferation rate at day six to obtain the proliferation rate time-course:

$$r(t) = \frac{mi(t)}{mi(day\ 6)} r(day\ 6),$$

where *r(t)* is the proliferation rate at time *t*, and *mi* is the mitotic index. The mitotic index in the high-proliferation zone was estimated as described in (*Rodrigo Albors et al., 2015*).

## Axolotl spinal cord electroporation

Axolotl larvae (2 cm snout-to-tail) were electroporated with a dual fluorescent reporter plasmid (cytoplasmic eGFP and nuclear Cherry). Cells were electroporated by cutting the tail of 2 cm-long larval axolotls and inserting a DNA-filled electrode into the spinal cord (*Echeverri and Tanaka, 2003*). To transfect DNA into only a few cells, optimum electroporation conditions were three pulses of 50 V, 200 Hz and a length of 100 ms, applied using an SD9 Stimulator (Grass Telefactor, West Warwick, RI).

## In vivo imaging of labeled cells in the spinal cord

Axolotls with sparsely labelled cells in the spinal cord were amputated, leaving cells at different distances from the amputation plane. Regenerating axolotls were anaesthetized and imaged every 1–2 days by placing them on a cover slip. Labelled cells were imaged using a Zeiss Apotome A1 microscope.

## Clone tracking

The distance between the amputation plane and the anterior border of a clone was measured manually in each image using AxioVision microscopy software (RRID:SCR_002677). Representative images of one axolotl showing a clone at different distances from the amputation plane during regeneration time are shown in *Figure 2I*. All the individual images are in *Rost et al., 2016c*.

## Clone velocity

To estimate the mean velocity of clones at different spatial positions, the space along the AP axis was subdivided into 800 µm bins. For each clone trajectory, the position measurements were grouped according to these bins. Groups containing less than two measurements were excluded. The average clone velocity for each group was estimated with linear regression. Then, the mean and standard deviation of the velocity of all the clones in a bin was calculated (see 'clone_velocities' in *Rost et al., 2016a*).

## Estimation of the total number of quiescent cells in the high-proliferation zone

The total number of quiescent cells in the high-proliferation zone, $N_q(\mathrm{t})$, was estimated by $N_q(t) = N_q^s \cdot L(t)/l_{cell}$ where $N_q^s$ is the mean number of SOX2$^+$/PCNA$^-$ cells per cross section, $l_{cell}$ is the mean AP cell length, and $L(t)$ is the outgrowth time-course.

## Mechanistic model of spinal cord outgrowth

To simultaneously evaluate the importance of cell proliferation, cell influx and activation of quiescent cells in the outgrowth of the spinal cord we performed a data-driven modeling approach (*Greulich and Simons, 2016*; *Rué and Martinez Arias, 2015*; *Oates et al., 2009*). This approach allows to establish causal relationship between the individually quantified cellular processes and it has been previously employed to unravel the stem cell dynamics during spinal cord development in chick and mouse (*Kicheva et al., 2014*). Although less frequent so far, modeling is more and more being used in the regeneration arena (*Durant et al., 2016*; for an overview see *Chara et al., 2014*). In this study, we model the number of proliferative, $N_p(t)$, and quiescent cells, $N_q(t)$, in the high-proliferation zone at time $t$ by the following ordinary differential equations (*Figure 3A*):

$$\frac{dN_p(t)}{dt} = \overbrace{r(t)N_p(t)}^{\text{cell divisions}} + \overbrace{kN_q(t)}^{\text{activation}} + \overbrace{\frac{N_p(t)}{N_p(t)+N_q(t)}v\rho}^{\text{influx}} \ , \qquad N_p(t=0)=N_p^0, \tag{3}$$

$$\frac{dN_q(t)}{dt} = \quad - \quad kN_q(t) \quad + \quad \frac{N_q(t)}{N_p(t)+N_q(t)}v\rho \ , \qquad N_q(t=0)=N_q^0. \tag{4}$$

where $N_p^0$ and $N_q^0$ are the initial cell numbers in this zone, $r(t)$ is the proliferation rate at time $t$, $v$ is the velocity of cells 800 µm anterior to the amputation plane, $\rho$ is the density of neural stem cells along the AP axis and $k$ is the quiescent cell activation rate. The factors $N_{p/q}(t)/ (N_p(t)+ N_q(t))$ ensure that the influx of cells into the high-proliferation zone does not alter the growth fraction. As the density is constant one can write

$$\rho \cdot (L(t)+L_0) = N_p(t)+N_q(t), \tag{5}$$

where $L(t)$ is the outgrowth posterior to the amputation plane and $L_0$ = 800 µm is the high-proliferation zone length at $t$ = 0. Using this relation and the definition of the growth fraction $GF(t)$,

$$GF(t) = \frac{N_p(t)}{N_p(t)+N_q(t)}, \tag{6}$$

the cell number model was reformulated as a model for outgrowth and growth fraction (see Results, *Equations 1 and 2*).

The assumption that the population mean model parameters can be used to estimate the population mean outgrowth time-course was used when simulating the model and interpreting results. The confidence intervals of the model prediction were estimated with a Monte Carlo approach using bootstrapping with a case resampling scheme (100,000 iterations). In each iteration, we case-resampled the cell count data, the BrdU incorporation data and the clone trajectory data, and calculated the proliferation rate time-course, clone velocity at −800 µm and initial growth fraction from this resampled data as described above. Then, in each iteration, these bootstrapped parameter values were used to estimate the activation rate $k$ by fitting the model prediction of the growth fraction to the data (*Figure 3B*). The growth fraction measurement of day 8 was excluded from the fit because its precise value would only affect the model prediction after this day. Now, as all parameters were estimated, an outgrowth trajectory was calculated for each iteration. This ensemble of trajectories was used to calculate the confidence intervals of the model prediction (*Figure 3C*). The same approach was used for the model scenarios with individual cellular mechanisms turned off (*Figure 3D–G*). The source code is available in the 'lg_model' in *Rost et al., 2016a*.

## Validation of a model prediction against an experimental dataset

Control animals by *Fei et al. (2014)* showed less regenerative outgrowth than our 'normally' regenerating animals. This could be either due to their control CRISPR treatment or due to their reduced feeding. To account for the reduced growth, we assumed that all cellular mechanisms maintain the same relative contribution in Fei and colleagues' control as they have in normal regeneration. This assumption allowed linear rescaling of the outgrowth dataset from Fei and colleagues to match our 'normal' outgrowth dataset (*Figure 3—figure supplement 2A*, 'lg_model' in *Rost et al., 2016a*). We also assumed that *Sox2*-knockout only affects the acceleration of the cell cycle but that all other cellular mechanisms remain unaffected (i.e. compensatory mechanisms are not considered). Fewer neural stem cells make up the circumference of *Sox2*-knockout spinal cords (*Fei et al., 2014*). Assuming that the AP cell length is unchanged this means that cell density is decreased in this condition. Therefore, we corrected the outgrowth for the *Sox2*-knockout dataset to a density corrected outgrowth by $L_{corr} = N_S^{Sox2^{ko}}/N_S^{control} \cdot L$, where $L_{corr}$ is the density corrected outgrowth, $L$ is the measured outgrowth in the *Sox2*-knockout dataset and $N_S^{Sox2^{ko}}$ and $N_S^{control}$ are the mean number of neural stem cells per cross section in the *Sox2*-knockout and control condition, respectively (*Figure 3—figure supplement 2B*, 'lg_model' in *Rost et al., 2016a*).

## Coordinate system

Time starts with the event of amputation. For spatial positions along the AP axis of the spinal cord, the amputation plane defines 0; positive values refer to positions posterior to the amputation plane, in regenerated tissue; negative values refer to positions anterior to the amputation plane. In all images, anterior is to the left.

## Statistics and computational tools

If not stated otherwise, measurements are reported as mean ± standard error of the mean. In the figures * denotes $p<0.05$ and ** denotes $p<0.01$ for the respective test as indicated in the figure caption.

Image analysis was performed with Fiji (*Schindelin et al., 2012*) and AxioVision Microscopy software (Zeiss). Data analysis was performed using the python modules bokeh (http://bokeh.pydata.org), iminuit (http://github.com/iminuit/iminuit), ipycache (http://github.com/rossant/ipycache), Jupyter Notebook (http://jupyter.org/), matplotlib (*Hunter, 2007*), numba (http://numba.pydata.org/), pandas (*McKinney, 2010*), probfit (http://github.com/iminuit/probfit), pymc (*Patil et al., 2010*), SciPy (*Jones et al., 2001*) and uncertainties (http://pythonhosted.org/uncertainties/).

## Supplementary notebooks

Jupyter Notebooks containing the source code for all computations performed together with the data and referred to as *Rost et al., 2016a* in this work can be found at https://doi.org/10.5281/zenodo.160333.

## Acknowledgements

We are grateful to Beate Gruhl, Sabine Mögel, Anja Wagner, and Heino Andreas for outstanding axolotl care. We thank Walter de Back, Nuno Barros, Emanuel Cura Costa, Ji-Feng Fei, Keisuke Ishihara, Jörn Starruß and Anja Voß-Böhme for helpful discussions.

## Additional information

### Funding

| Funder | Grant reference number | Author |
| --- | --- | --- |
| Human Frontier Science Program | RGP0016/2010 | Fabian Rost<br>Aida Rodrigo Albors<br>Lutz Brusch<br>Andreas Deutsch<br>Elly M Tanaka<br>Osvaldo Chara |

| | | |
|---|---|---|
| DIGS-BB Program | PhD Fellowship | Aida Rodrigo Albors |
| Bundesministerium für Bildung und Forschung | 0316169A | Lutz Brusch |
| Deutsche Forschungsge-meinschaft | DFG-274/2-3/SFB655 | Elly M Tanaka |
| Agencia Nacional de Promo-ción Científica y Tecnológica | PICT-2014-3469 | Osvaldo Chara |

The funders had no role in study design, data collection and interpretation, or the decision to submit the work for publication.

### Author contributions
FR, EMT, OC, Conception and design, Analysis and interpretation of data, Drafting or revising the article; ARA, Conception and design, Acquisition of data, Analysis and interpretation of data, Drafting or revising the article; VM, Acquisition of data, Drafting or revising the article; LB, AD, Analysis and interpretation of data, Drafting or revising the article

### Author ORCIDs
Fabian Rost, http://orcid.org/0000-0001-6466-2589
Aida Rodrigo Albors, http://orcid.org/0000-0002-9573-2639
Osvaldo Chara, http://orcid.org/0000-0002-0868-2507

### Ethics
Animal experimentation: The axolotl animal work was performed under permission granted in animal license number DD24-9168.11-1/2012-13 conferred by the Animal Welfare Commission of the State of Saxony, Germany (Landesdirektion, Sachsen).

## Additional files

### Major datasets
The following datasets were generated:

| Author(s) | Year | Dataset title | Dataset URL | Database, license, and accessibility information |
|---|---|---|---|---|
| Fabian Rost, Aida Rodrigo Albors, Vladimir Mazurov, Lutz Brusch, Andreas Deutsch, Elly M Tanaka, Osvaldo Chara | 2016 | Accelerated cell divisions drive the outgrowth of the regenerating spinal cord in axolotls - Supplementary notebooks - v1.0 | https://doi.org/10.5281/zenodo.160333 | Publicly available at Zenodo (https://zenodo.org/) |
| Fabian Rost, Aida Rodrigo Albors, Vladimir Mazurov, Lutz Brusch, Andreas Deutsch, Elly M Tanaka, Osvaldo Chara | 2016 | Accelerated cell divisions drive the outgrowth of the regenerating spinal cord in axolotl - Supplementary file 1 | http://dx.doi.org/10.5281/zenodo.59817 | Publicly available at Zenodo (https://zenodo.org/) |
| Fabian Rost, Aida Rodrigo Albors, Vladimir Mazurov, Lutz Brusch, Andreas Deutsch, Elly M Tanaka, Osvaldo Chara | 2016 | Accelerated cell divisions drive the outgrowth of the regenerating spinal cord in axolotl - Supplementary file 2 | http://dx.doi.org/10.5281/zenodo.59824 | Publicly available at Zenodo (https://zenodo.org/) |

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
