## [Decision Letter]

Thank you for submitting your Research Advance article "Accelerated cell divisions drive the outgrowth of the regenerating spinal cord in axolotls" for consideration by *eLife*. Your article has been reviewed by two peer reviewers, and the evaluation has been overseen by Marianne Bronner as the Senior Editor and Reviewing Editor. The reviewers have opted to remain anonymous.

The reviewers have discussed the reviews with one another and the Reviewing Editor has drafted this decision to help you prepare a revised submission.

Summary:

This Research Advance paper is a follow up of a previous paper from the same group in which they demonstrated that neural stem cells in the axolotl spinal cord re-establish a planar cell polarity signaling during spinal cord regeneration. In this paper, they describe in more detail the cell proliferative response during regeneration and identify a dynamic high-proliferation zone in the regenerating spinal cord. The authors reasoned that this increased proliferation could be due to 1) changes in the cell proliferation rate, 2) activation of quiescent cells, and 3) cell influx into the regenerating spinal cord.

Essential revisions:

The authors should soften their statement about acceleration of cell cycle as being "necessary and sufficient" to drive regeneration since these are predictions based on mathematical modeling. The major concern about this paper is the lack of validation of the model predictions, especially since the authors use the model to predict the data from which the model was derived. The predictive value of the model would have been strengthened by applying it to an independent dataset. Any one of the following experiments should be included to strengthen the paper:

1) Reduce cell cycle in proliferating stem cells and demonstrate that regeneration is delayed to a similar extent as predicted by the model. The authors cite previous *Sox2* KO study in support of their findings, but perhaps they can use that dataset to test the robustness of their model.

2) Prevent the influx of cells into the regenerating spinal cord and demonstrate that it has negligible effects on regeneration.

3) Prevent the activation of quiescent stem cells and demonstrate that it has negligible effects on regeneration. It is possible that one of the three mechanisms may compensate for the loss of the other. How does the model take this into account? This would be nice to include in the Discussion.

---

## [Author Response]

*Essential revisions:*

*The authors should soften their statement about acceleration of cell cycle as being "necessary and sufficient" to drive regeneration since these are predictions based on mathematical modeling. The major concern about this paper is the lack of validation of the model predictions, especially since the authors use the model to predict the data from which the model was derived. The predictive value of the model would have been strengthened by applying it to an independent dataset. Any one of the following experiments should be included to strengthen the paper:*

*1) Reduce cell cycle in proliferating stem cells and demonstrate that regeneration is delayed to a similar extent as predicted by the model. The authors cite previous Sox2 KO study in support of their findings, but perhaps they can use that dataset to test the robustness of their model.*

*2) Prevent the influx of cells into the regenerating spinal cord and demonstrate that it has negligible effects on regeneration.*

*3) Prevent the activation of quiescent stem cells and demonstrate that it has negligible effects on regeneration. It is possible that one of the three mechanisms may compensate for the loss of the other. How does the model take this into account? This would be nice to include in the Discussion.*

We thank the reviewers for raising this concern. We would like to mention that we devised the model to understand the contribution of individual cellular mechanisms to the normal regenerative spinal cord outgrowth. To do so, we (i) measured spinal cord outgrowth during regeneration (at the tissue level) and (ii) quantified different cellular mechanisms that underlie this process (at the cell level). We then used the cell-level datasets to build the model, which we used to predict tissue outgrowth. These tissue outgrowth predictions were tested against the experimentally observed tissue outgrowth dataset. Thus, this tissue-level dataset was not used to derive the model but rather to validate it. Next, to assess the contribution of each cellular mechanism to normal regenerative spinal cord outgrowth, we switched them off one by one in silico. Here, we agree with the reviewers that these predictions would be strengthened when tested against an independent experimental dataset in which one of the cellular mechanisms is blocked. However, a motivation to develop the model (and a limitation to validate its predictions in perturbed conditions) is that we know little about how these cellular mechanisms are regulated in the axolotl – except for the acceleration of the cell cycle. As mentioned in the first version of this paper and cited by the reviewers, knocking out *Sox2* hinders the acceleration of the cell cycle during spinal cord regeneration in axolotls (PMCID: PMC4266004). In this revised version of the manuscript, we have followed the reviewers’ suggestion to validate the model on the *Sox2*-knockout dataset from Fei and colleagues (PMCID: PMC4266004, Figure 4). We now show that our model can explain the reduced regenerative outgrowth in *Sox2*-knockout spinal cords when the acceleration of the cell cycle is switched off. Please refer to Figure 3—figure supplement 2; subsection “Cell proliferation drives the outgrowth of the regenerating spinal cord”, last paragraph; Discussion, fourth paragraph subsection “Validation of a model prediction against an experimental dataset” in Results, Discussion and Materials and methods, respectively.

In addition and as suggested by the reviewers, we have reworded the statement that the acceleration of the cell cycle is “necessary and sufficient” to drive spinal cord outgrowth. We now state instead that the acceleration of the cell cycle is the major driver of regenerative spinal cord outgrowth (see legend of Figure 4 and lines Abstract; Introduction, third paragraph; subsection “Cell proliferation drives the outgrowth of the regenerating spinal cord”, third paragraph; Discussion, third paragraph).

“It is possible that one of the three mechanisms may compensate for the loss of the other. How does the model take this into account? This would be nice to include in the discussion.”

This is certainly a possibility, but our model does not consider compensatory mechanisms when one of the mechanisms is perturbed. However, the fact that our model, without considering compensatory mechanisms, can predict the reduced outgrowth when the acceleration of cell proliferation is blocked (as described above) suggests that there is not massive compensation by one of the other mechanisms during regeneration. We now comment on this in the Discussion, fourth paragraph.